# A plant tendril mimic soft actuator with phototunable bending and chiral twisting motion modes

Meng Wang[1], Bao-Ping Lin[1] & Hong Yang[1]

In nature, plant tendrils can produce two fundamental motion modes, bending and chiral twisting (helical curling) distortions, under the stimuli of sunlight, humidity, wetting or other atmospheric conditions. To date, many artificial plant-like mechanical machines have been developed. Although some previously reported materials could realize bending or chiral twisting through tailoring the samples into various ribbons along different orientations, each single ribbon could execute only one deformation mode. The challenging task is how to endow one individual plant tendril mimic material with two different, fully tunable and reversible motion modes (bending and chiral twisting). Here we show a dual-layer, dual-composition polysiloxane-based liquid crystal soft actuator strategy to synthesize a plant tendril mimic material capable of performing two different three-dimensional reversible transformations (bending versus chiral twisting) through modulation of the wavelength band of light stimuli (ultraviolet versus near-infrared). This material has broad application prospects in biomimetic control devices.

[1] School of Chemistry and Chemical Engineering, Jiangsu Province Hi-Tech Key Laboratory for Bio-medical Research, Jiangsu Optoelectronic Functional Materials and Engineering Laboratory, State Key Laboratory of Bioelectronics, Southeast University, Nanjing, Jiangsu Province 211189, China. Correspondence and requests for materials should be addressed to H.Y. (email: yangh@seu.edu.cn).

In nature, plant tendrils can produce two fundamental motion modes, bending and chiral twisting (helical curling) distortions, under the stimuli of sunlight, humidity, wetting and other atmospheric conditions[1–4]. These motions are induced by the release of stored elastic energies, which derive from their non-uniform internal structures possessing different oriented layers, rigidities, expansion or swelling properties. Learned from these biological mechanisms, many artificial plant-like devices have been developed. For example, Smalyukh and colleagues[5] recently reported a method of controlling complex shapes (knot, bend, twist and so on) of tube-like polymer particles in liquid crystal through varying surface boundary conditions, to generate topological defects. Besides these tendril-like materials with fixed bend/twist shapes, stimuli-responsive soft actuators capable of performing reversible bending or chiral twisting motions have potential applications in robotic technology and thus have attracted more and more scientific attention.

To fabricate such stimuli-responsive soft actuators, the essential element is to understand and further mimic bending and chiral twisting motions. As it is well known, bending is the most common three-dimensional (3D) deformation mode, which results from inhomogeneous expansion/shrinking behaviours[6] of materials and has been realized in hydrogels[7], polyelectrolytes[8], shape-memory polymers[9] and so on. In particular, azobenzene-incorporated liquid crystalline elastomer (LCE) materials, can efficiently perform reversible bending deformations relying on the cis–trans isomerization of azobenzene chromophores under ultraviolet irradiation[10–15]. Very recently, an incredible sunlight-driven continuous oscillatory bending motion has been realized in a fluorinated azobenzene-embedded LCE material[16].

Chiral twisting, as a more complicated 3D deformation motion, is more challenging to manufacture than bending mode and has attracted intense attention. The most dominant strategy is to build bimorph sheets with different properties of the top and bottom layers/surfaces. For example, Sharon and colleagues[17] glued together two uniaxially stretched latex sheets in perpendicular directions, and from the composite material cut out elongated strips, which could mimic plants' helical curling motions once swelled in water. Taking advantages of LCE's anisotropic orientational orders and rubbery elastic properties[18–27], Urayama and colleagues[28], White and colleagues[29], and Katsonis and colleagues[30], respectively, synthesized thermal-responsive and photo-responsive twist-nematic LCE spiral ribbons using the classical LC-cell-alignment protocol[11,31]. Another very interesting strategy was demonstrated by Kumacheva and colleagues[32,33], and Broer and colleagues[34] recently. Instead of multi-layer build-up, Kumacheva and colleagues[32,33] fabricated a single-layer hydrogel sheet with periodic stripes of different compositions, which exhibited different swelling ratios and elastic moduli under external stimuli, to generate helices. Broer and colleagues[34] developed a versatile method for preparing a variety of humidity-responsive actuators based on a single-layer sheet comprising a hydrogen-bonded, uniaxially aligned LCE network.

However, all the above soft actuator materials were not able to fully mimic a plant tendril, which can realize not only bending but also chiral twisting (left-handed and right-handed) in one single piece, as shown in Fig. 1a. In another word, although some materials could achieve bending and helical curling through tailoring the samples into various ribbons along different orientations[17,28–30], each single ribbon could execute only one deformation mode. Herein an interesting question arises: is it possible to synthesize a real plant tendril mimic material capable of performing tunable, reversible bending and chiral twisting motions under two different external stimuli? This challenging task is the objective of this work. Herein we describe a dual-layer, dual-composition polysiloxane-based LCE strategy, to fabricate a plant tendril mimic material capable of performing two different 3D transformations (bending and chiral twisting) through modulation of the wavelength band of light stimuli.

## Results

**Design and synthesis.** Inspired by all the above landmark works, we rely on the photo-deformable and stimuli-responsive 3D liquid crystal soft actuator system[35–39], design and synthesize a plant tendril mimic material comprised of a dual-layer, dual-composition polysiloxane-based LCE structure, as schematically illustrated in Fig. 1b,c. The top layer, also assigned as the main skeleton, possesses a uniaxially aligned LCE matrix incorporated with azobenzene chromophores and a near-infrared absorbing dye, so that the main skeleton of this material can execute bending under ultraviolet stimulus and shrinking under near-infrared stimulus[40] because of the azobenzene cis–trans isomerization effect and the photo-thermal heating effect[21,41–48], which would induce the LC-to-isotropic phase transition, respectively, whereas the bottom monodomain LCE layer, which was obliquely glued on the main skeleton, has no azobenzene moieties but the near-infrared dye, so that it can only respond to near-infrared stimulus, and contributed a twisting power for the whole material to helically curl, because the shrinkage directions of the top and bottom layer are tilted to each other. The different overlapped angles between the top and bottom layers (45° or −45°) can force the material to perform right-handed or left-handed helical curling. Overall, such a plant tendril mimic material will bend under ultraviolet illumination and helically curl towards near-infrared irradiation. Most importantly, these two motions of this LCE soft actuator are fully reversible.

As shown in Fig. 1b, the composition of the top LCE layer (PMHS-AZO46-MBB/YHD796 composite, Formula 1) included a polymer backbone polymethylhydrosiloxane (PMHS), a cross-linker 1,4-bis-undec-10-enyloxy-benzene[49] (11UB, 9 mol%), a classical nematic monomer 4-pent-4-enyloxy-benzoic acid 4-butoxy-phenyl ester (MBB, 68 mol%), an azobenzene mesogenic monomer (4-butoxy-phenyl)-(4-hex-5-enyloxy-phenyl)-diazene (AZO46, 23 mol%), a near-infrared absorbing dye YHD796 (ref. 50) (0.5 wt%) and a platinum catalyst dichloro(cycloocta-1,5-diene) platinum (II) (Pt(COD)Cl$_2$). The composition of the bottom LCE layer (PMHS-MBB/YHD796 composite, Formula 2) was similar to that of the top layer, except that only MBB was used as the mesogenic monomer. The two-step hydrosilylation crosslinking process by Küpfer and Finkelmann[51,52] coupled with a uniaxial stretching technique[53] was then applied to fabricate the bilayer LCE material, as demonstrated in Fig. 1c. Specifically, two mixtures (Formula 1 and 2) composed of the above reagents dissolved in toluene were cast into two polytetrafluoroethylene (PTFE) rectangular moulds (2.0 cm long × 2.0 cm wide × 1.5 cm deep), respectively. After high-power ultrasonication for 5 min, the two PTFE moulds were heated in an oven at 60 °C for 4 h to carry out a partial hydrosilylation crosslinking process. After cooling to room temperature, the two LCE samples were removed from the PTFE moulds, dried overnight and sliced into stripes. The stripe films were slowly uniaxially stretched to 140∼160% of the original lengths. Then, the pre-crosslinked PMHS-AZO46-MBB/YHD796 composite film was obliquely placed on the top of the pre-crosslinked PMHS-MBB/YHD796 composite film with a crossed angle of either 45° or −45°. The fixed dual-layer LCE film was heated at 60 °C in an oven for 72 h to accomplish

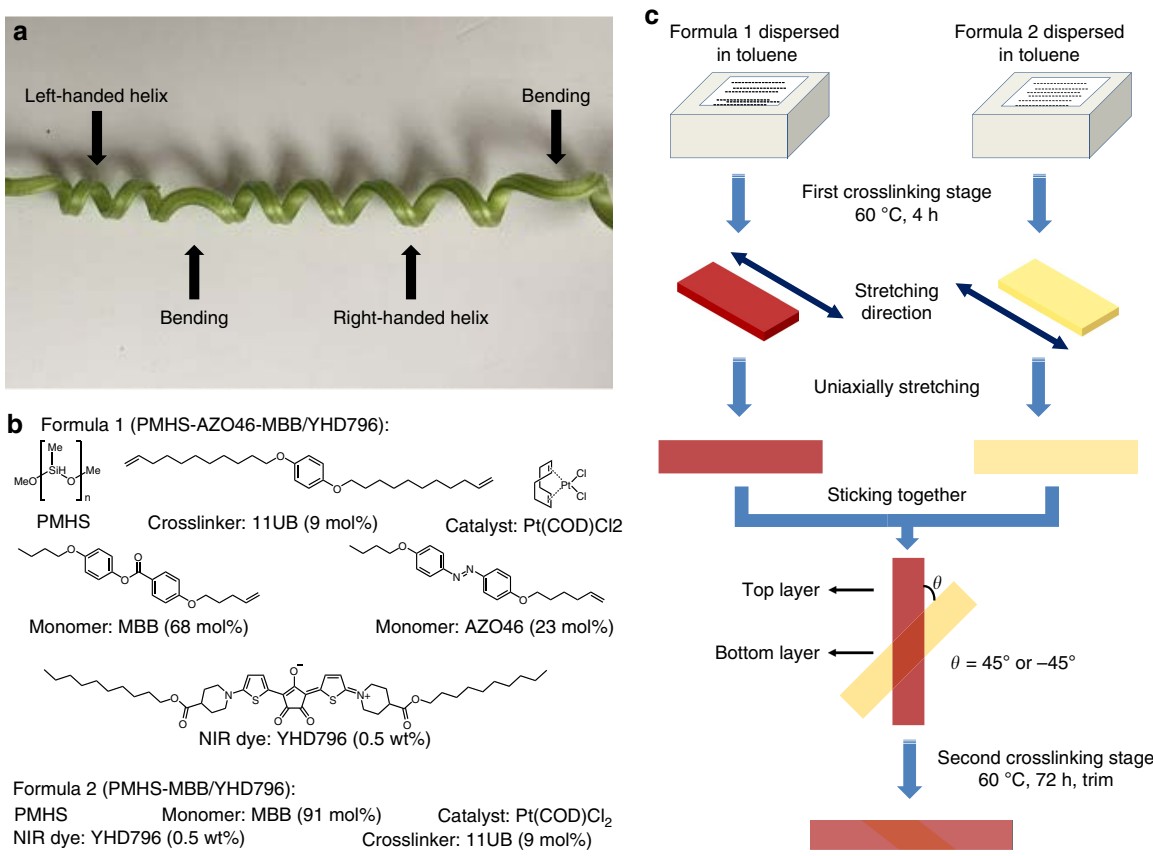

**Figure 1 | Design and synthesis of dual-layer polysiloxane-based liquid crystal soft actuators.** (**a**) The photo image of a cucumber plant tendril with bending and chiral twisting distortions. (**b**) The chemical compositions of PMHS-AZO46-MBB/YHD796 composite (Formula 1) and PMHS-MBB/YHD796 composite (Formula 2). (**c**) Schematic illustration of the preparation protocol of the dual-layer LCE ribbon material.

the full crosslinking procedure, during which the leftover unreacted vinyl groups and Si-H groups on the interfaces of the two pre-crosslinked LCE samples would be covalently bonded together and consequently the two LCE layers would be spontaneously glued together. Finally, the dual-layer film was further trimmed along the stretching direction of the top layer, into ribbons with a dimension of *ca.* 2.0 cm long × 0.2 cm wide × 0.2 mm thick.

Compared with the previous methods for synthesizing helical curling materials[17,28–30,34], this preparation protocol has two technical advantages: first, unlike the classical LC-cell-alignment procedure, which usually prepared very thin films due to the limitation of cell thickness (*ca.* 20 ∼ 100 μm)[28–30,34], this approach can synthesize much thicker LCE materials capable of transmitting heavier loads. Second, taking advantage of Finkelmann[51,52] two-step crosslinking mechanism, the two different pre-crosslinked LCE samples could be spontaneously glued together during the second hydrosilylation crosslinking period, without using extra adhesives[17].

**Investigation of photoresponsive properties.** In Fig. 2a, we recorded the differential scanning calorimetry data of the LCE sample containing not only PMHS-AZO46-MBB/YHD796 composite but also PMHS-MBB/YHD796 composite, which had an enantiotropic nematic (N) phase. The glass transition ($T_g$) and the LC-to-isotropic phase transition temperatures ($T_{ni}$) of the bilayer LCE sample were approximately − 10 °C and 65 °C, whereas the clearing points of pure PMHS-MBB/YHD796 sample

and PMHS-AZO46-MBB/YHD796 sample were 57 °C and 67 °C, respectively (Supplementary Fig. 1). To investigate the optical absorption property of the bilayer LCE sample, the ultraviolet–visible spectra of YHD796, AZO46, PMHS-MBB/YHD796 (bottom layer) and PMHS-AZO46-MBB/YHD796 (top layer) were all merged in Fig. 2b for comparison. Both AZO46 and PMHS-AZO46-MBB/YHD796 samples dispersed in $CH_2Cl_2$ with a concentration of *ca.* $1.4 \times 10^{-3}$ mol l$^{-1}$ had one intense absorption peak in the ultraviolet region centred at 365 nm, whereas in the near-infrared region YHD796, PMHS-MBB/ YHD796 and PMHS-AZO46-MBB/YHD796 samples all showed a maximum absorption at 796 nm. Overall, it can be concluded that PMHS-AZO46-MBB/YHD796 composite film (top layer) could absorb photons in both the ultraviolet and near-infrared regions, whereas YHD796/PMHS-MBB composite sample (bottom layer) was only sensitive to near-infrared light.

Encouraged by the optical absorption results, we applied ultraviolet light and near-infrared light sources respectively, to investigate the photo-responsive actuation behaviours of the bilayer LCE ribbons at room temperature. As shown in Fig. 3a,b and Supplementary Movie 1,2, both the bilayer LCE ribbons ($\theta = 45°$ or $−45°$) could bend towards to the ultraviolet incoming direction, although the photoresponsive rates were modest (*ca.* 12 min). If the LCE ribbon was turned upside down and exposed to ultraviolet light (Fig. 3c and Supplementary Movie 3), the bending behaviour took place at the two edges of the film, whereas the centred two-layer-overlapped region remained motionless. As shown in Fig. 3d, the included angle α of line $l_1$ and line $l_2$ was measured at

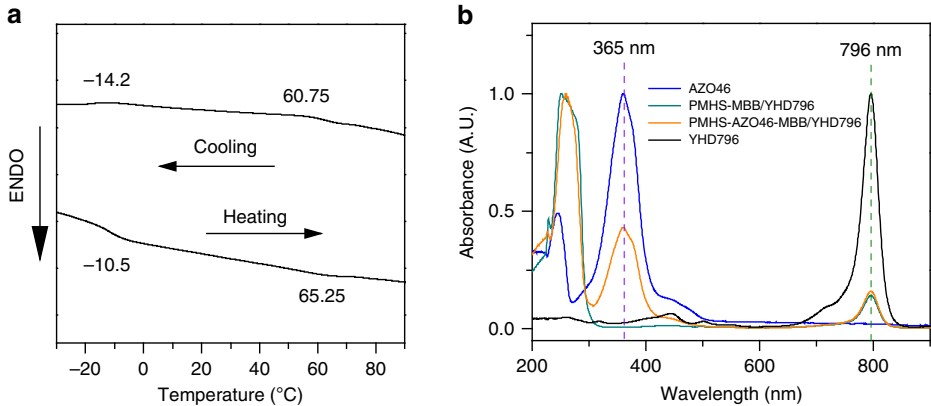

**Figure 2 | Thermal and optical absorption properties of the dual-layer LCE film.** (**a**) Differential scanning calorimetry curves of the LCE sample containing not only PMHS-AZO46-MBB/YHD796 composite but also PMHS-MBB/YHD796 composite. (**b**) Ultraviolet–vis spectra of YHD796, AZO46, PMHS-MBB/YHD796 and PMHS-AZO46-MBB/YHD796 composite films dispersed in $CH_2Cl_2$ with a concentration of $ca.$ $1.4 \times 10^{-3}$ mol l$^{-1}$.

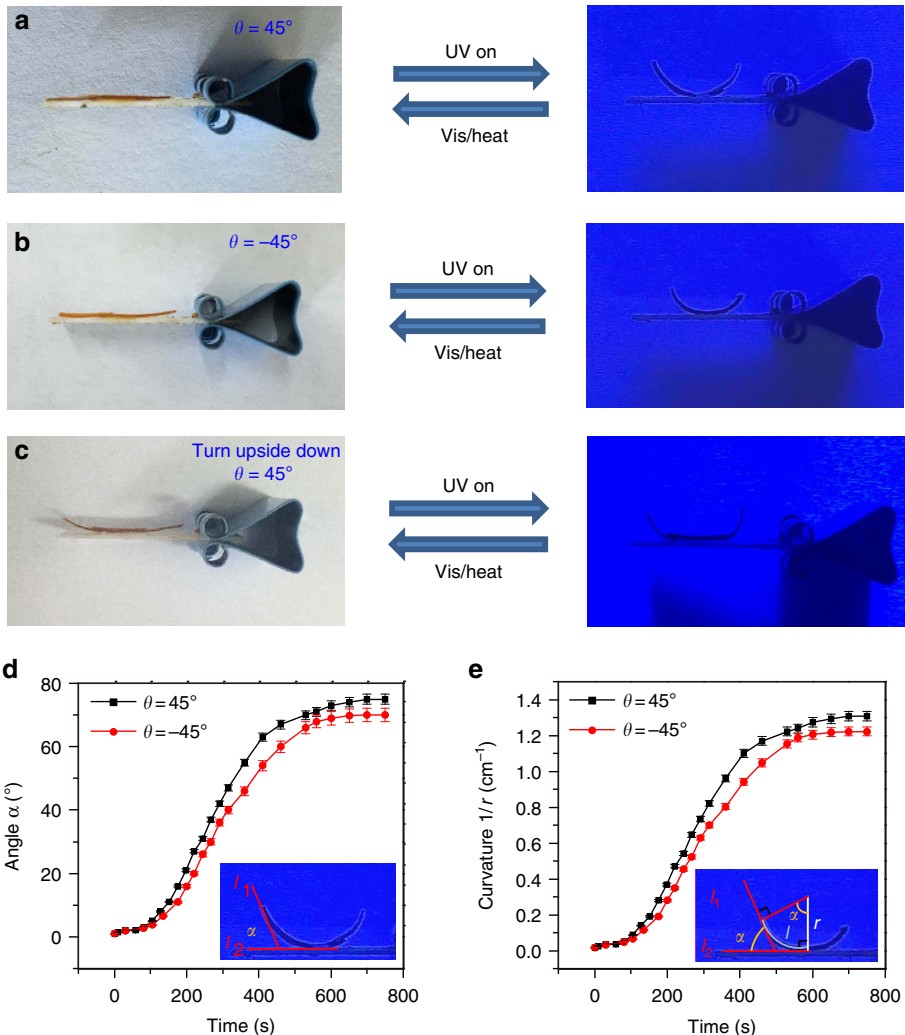

**Figure 3 | Ultraviolet-induced bending behaviour of two bilayer LCE ribbons.** The bilayer LCE ribbon with a (**a**) 45° or (**b**) −45° angle between the top and bottom layer was irradiated under 365 nm ultraviolet light. (**c**) The bilayer ribbon ($\theta = 45°$) was turned upside down and irradiated under 365 nm ultraviolet light. Supplementary Movies 1–3 show these scenarios in motion. (**d**) The included angle $\alpha$ vs ultraviolet illumination time diagram of the bilayer LCE ribbons. The error bars indicate the standard deviation of the measured angles. (**e**) Curvature ($1/r$) of the bilayer LCE ribbons as a function of ultraviolet illumination time. The error bars indicate the standard deviation of the bending curvature calculated from the included angle data.

different ultraviolet illumination time points, to show the photoresponsive rate of the bilayer LCE ribbons bending towards the ultraviolet light, where $l_1$ and $l_2$ are the tangent lines to the left endpoint and midpoint of the plane curve, respectively. The bending curvature ($1/r$) was further plotted against the ultraviolet illumination time (Fig. 3e),

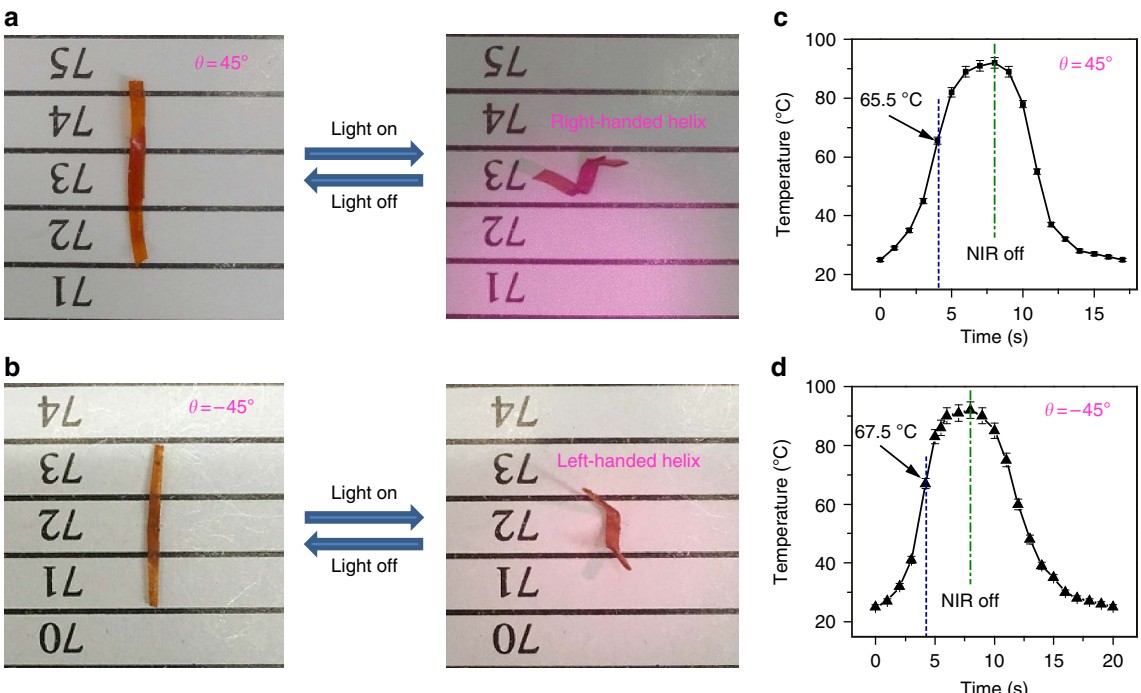

**Figure 4 | Near-infrared-induced chiral twisting behaviour of two bilayer LCE ribbons.** The bilayer LCE ribbon with a (**a**) 45° or (**b**) − 45° angle between the top and bottom layer was irradiated under an 808 nm near-infrared light for 8 s. Supplementary Movies 4 and 5 show these scenarios in motion. (**c**,**d**) Temperature versus near-infrared illumination time diagrams of the two bilayer LCE ribbons. The error bars shown in (**c**) and (**d**) represent the standard deviation of the measured surface temperature data of two bilayer LCE samples.

using the following equation: $1/r = (\alpha \div 180° \times \pi) \div l$, where $l$ is the length of the half arc. Under irradiation, the ribbons seemed not very photo-sensitive and remained almost motionless in the first 1.5 min, then started to perform a continuous bending behaviour with a roughly linear photoresponsive rate feature in the next 5.5 min and an exponential responsive kinetics later, and eventually reached the maximum deformation in ca. 12 min.

It is well known that the molar absorption coefficient of azobenzene is very high under 365 nm ultraviolet light. The absorption of ultraviolet photons promote electrons from the highest occupied molecular orbital ($\pi$-orbital) to the lowest unoccupied molecular orbital ($\pi^*$-orbital) of the azobenzene group, resulting in the isomerization of azobenzene from its thermodynamically stable *trans*-conformation to *cis*-conformation and the decrease of azobenzene's molecule length from ca. 10 to 5.6 Å. As previously explained in literatures[10,13,15], such a bending scenario was induced by the fact that the *cis–trans* isomerization extent of azobenzene chromophores would vary depending on the ultraviolet-penetration depth so that the top surface region of LCE sample would shrink much more than the bottom region. These different contraction ratios of the top and bottom sides of LCE sample eventually forced the macroscopic material to bend towards the ultraviolet source. When the film was turned upside down and exposed to ultraviolet light, the top non-azo-layer (PMHS-MBB/YHD796 layer) would prevent ultraviolet photons from reaching the bottom azo-layer (PMHS-AZO46-MBB/YHD796 layer), because of the limited penetration ability of ultraviolet light. Consequently, the two-layer-overlapped region of the LCE film kept motionless. Moreover, when performing visible light irradiation (a CEL-HXF300 xenon lamp, Output power: 18.6 W) or heating, the bent films would return to their original states, as shown in Supplementary Movies 1–3.

A near-infrared light source (centre wavelength: 808 ± 3 nm, output power: 8 W) was used to investigate the near-infrared-triggered twisting actuation of the bilayer LCE ribbons. As shown in Fig. 4a,b and Supplementary Movies 4 and 5, the ribbons could curl into helical configurations in 8 s. After removing near-infrared source, the bilayer ribbons fully recover their original shapes. A thermal imager (FLUKE Ti90) was used to record the surface temperature changes of two bilayer LCE ribbons under the near-infrared-light illumination to further examine the curling behaviour of these two LCE ribbons. As shown in Fig. 4c,d, the temperature of two bilayer films rose from room temperature to ca. 65.5 ∼ 67.5 °C in 4 s, which was higher than their $T_{ni}$ (65.3 °C), and eventually reached to ca. 90 °C in 8 s. When near-infrared-light was removed, the ribbons' temperature quickly decreased to 30 °C in the next 10 s.

Different from the equal bending behaviour of two ribbons stimulated by ultraviolet light, the two bilayer LCE ribbons under near-infrared irradiation presented right-handed and left-handed helical curling motions, respectively, depending on the overlapped angle ($\theta = 45°$ or − 45°) between the top and bottom layer. During near-infrared irradiation, both the top and bottom layers of the bilayer LCE samples would be heated to above their clearing points because of the photo-thermal conversion effect of the embedded near-infrared dye YHD796. Consequently, the two layers tended to shrink along their own alignment orientations, which were tilted to each other with a 45° or − 45° angle, as illustrated in Supplementary Fig. 2. Such a non-uniform shrinkages created an incompatibility in the two-layer-overlapped region. Along each of their own alignment directions, the top and bottom layers had different contraction ratios, which resulted in two bending tendencies along the two alignment directions. The vector sum of the two bending deformations made an inclined angle with the long axis of the LCE ribbon, and eventually generated a chiral twisting power to force the macroscopically flat ribbon to curl in either right-handed or left-handed manner.

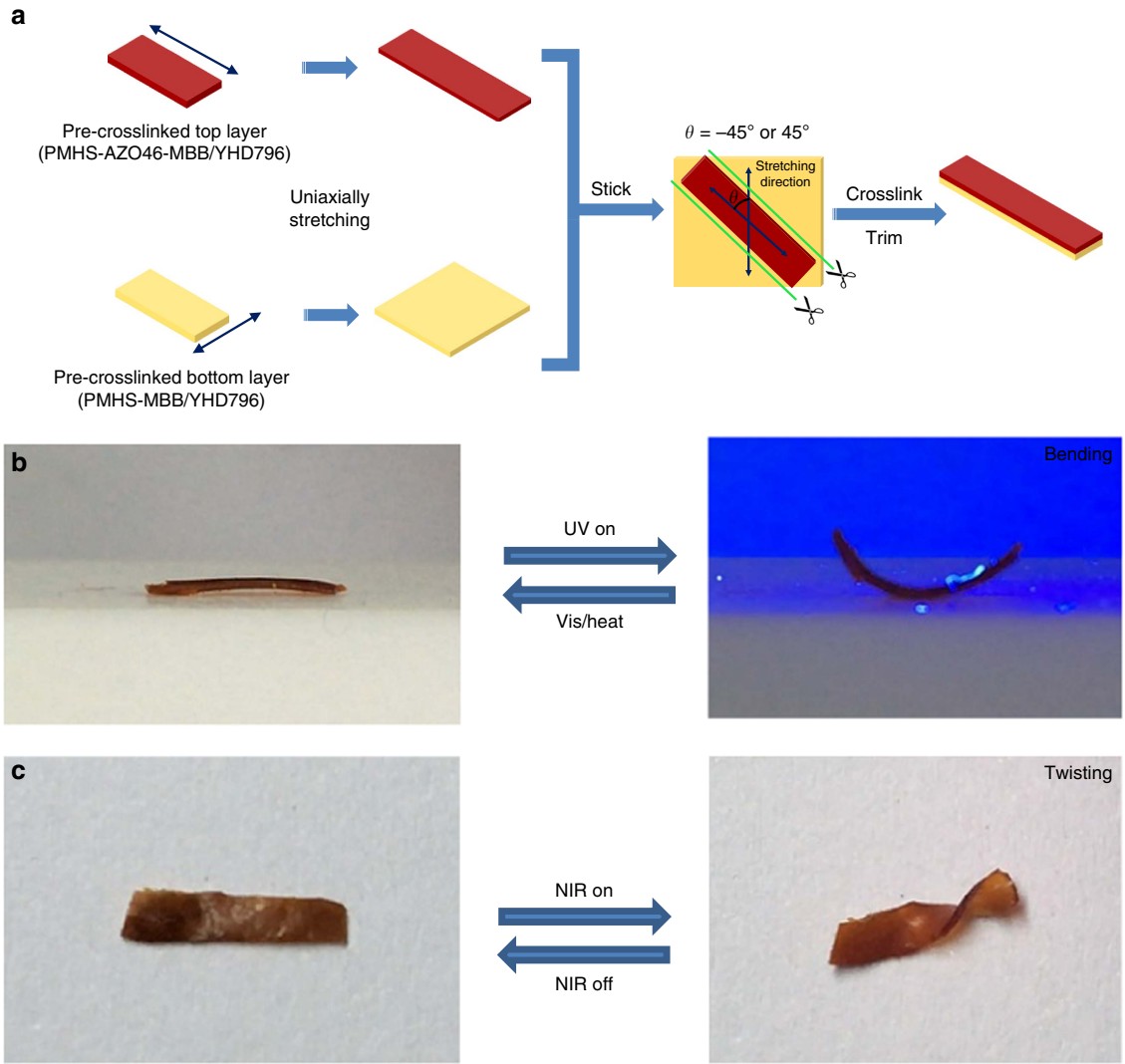

**Figure 5 | ultraviolet and near-infrared photoresponsive behaviours of a same-sized-bilayer LCE ribbon.** (**a**) Schematic illustration of the preparation protocol of a same-sized-bilayer LCE ribbon material whose top layer was of the same size as the bottom layer. The bilayer LCE ribbon with a − 45° angle between the top and bottom layer was irradiated under (**b**) 365 nm ultraviolet light and (**c**) an 808 nm near-infrared light, respectively. Supplementary Movies 6 and 7 show these scenarios in motion.

In addition to the above samples, which did curl selectively in the two-layer-overlapped regions, we further prepared an LCE actuator whose top layer was of the same size as the bottom layer, as schematically illustrated in Fig. 5a. Such a same-sized-bilayer sample could not only bend under ultraviolet irradiation (Fig. 5b) but also curl entirely under near-infrared illumination (Fig. 5c). Supplementary Movies 6 and 7 show the reversible ultraviolet-induced bending and near-infrared-induced curling behaviours, respectively.

## Discussion

Scientists have developed many elegant alignment methods to help stimuli-responsive materials construct hierarchical structures to mimic organisms' complex shape deformations. Here we demonstrate that through varying the chemical compositions of some hierarchical layers of soft actuators, these chemically different layers are capable of responding to different stimuli and the macroscopic materials might consequently be able to perform multiple shape deformations, which can be efficiently tuned by these different stimuli.

Moreover, this polysiloxane-based LCE system exhibits one extraordinary advantage in building multilayer hierarchical structures. Thanks to the wide adaptability and a moderate reaction rate of hydrosilylation reaction, the interface reactions between two different pre-crosslinked LCE polysiloxane substrates are readily allowed to covalently bond them together, which means that by employing the concept of this 'proof-of-idea' work, more complicated LCE actuators comprising two, three and four or even more layers with different shapes, different molecular orientations and different chemical compositions can also be prepared. We hope that these findings can pave the way for developing multi-stimuli responsive materials.

In conclusion, we developed a dual-layer, dual-composition polysiloxane-based LCE strategy to mimic an individual plant tendril, which could perform not only bending but also chiral twisting (left-handed and right-handed). The fundamental logic of this design is to make good use of the gap between the first pre-crosslinking and second full-crosslinking stages, to build up a multi-layer, multi-orientation, multi-composition LCE structure, to achieve the desired multi-stimuli responsive function. These soft actuator materials are capable of performing

two different reversible 3D transformations (bending versus chiral twisting) under irradiations of two light sources with different wavelength ranges (ultraviolet versus near-infrared), which might have potential applications in control devices and biomimetic devices, and so on.

## Methods

**General considerations.** All the starting reagents and instrumentation are described in Supplementary Methods. Detailed synthetic procedures and characterizations of small molecules (YHD796, MBB and AZO46) are included in Supplementary Figs 3–15 and Supplementary Methods. Seven short movies showing the photo-stimulated motions of LCE ribbons are recorded in Supplementary Movies 1–7. Specifically, Supplementary Movie 1 presents a side view of a bilayer LCE ribbon ($\theta = 45°$) under ultraviolet (365 nm) illumination (Video playback rate: 30X). Supplementary Movie 2 presents a side view of a bilayer LCE ribbon ($\theta = -45°$) under ultraviolet (365 nm) illumination (Video playback rate: 30X). Supplementary Movie 3 presents a side view of an upside down bilayer LCE ribbon ($\theta = 45°$) under ultraviolet (365 nm) illumination (Video playback rate: 30X). Supplementary Movie 4 presents a top view of a bilayer LCE ribbon ($\theta = 45°$) under near-infrared (808 nm) illumination (Video playback rate: 1X). Supplementary Movie 5 presents a top view of a bilayer LCE ribbon ($\theta = -45°$) under near-infrared (808 nm) illumination (Video playback rate: 1X). Supplementary Movie 6 presents a side view of a same-sized-bilayer LCE ribbon ($\theta = -45°$) under ultraviolet (365 nm) illumination (Video playback rate: 30X). Supplementary Movie 7 presents a top view of a same-sized-bilayer LCE ribbon ($\theta = -45°$) under near-infrared (808 nm) illumination (Video playback rate: 1X).

**Preparation of a pre-crosslinked PMHS-MBB/YHD796 film.** PMHS (24.0 mg, 0.400 mmol Si-H groups), MBB (104.0 mg, 0.349 mmol), 11UB (13.6 mg, 0.033 mmol) and YHD796 (0.7 mg, 0.5 wt%) were dissolved in 2 ml of toluene, which was then ultrasonicated for 5 min, to ensure a homogeneous dispersion. The mixture solution was cast into a PTFE rectangular mould (2.0 cm long × 2.0 cm wide × 1.5 cm deep). After adding 80 μl of the pre-catalyst solution (0.025 g of Pt(COD)Cl$_2$ was dissolved in 20 mL of CH$_2$Cl$_2$) into the above mixture, the PTFE mould was ultrasonicated for 5 min, to remove the bubbles in the mixture solution, and then heated in an oven at 60 °C for 4 h, to accomplish the first cross-linking stage. After cooling to room temperature, the LCE sample was carefully removed from the PTFE mould with the help of hexanes and then immediately cut into a strip (2.0 cm long × 0.5 cm wide, the thickness was ca. 0.1 mm), which was uniaxially stretched to ca. 140 ∼ 160% of the original length, fixed by using tapes and dried at room temperature.

**Preparation of a pre-crosslinked PMHS-AZO46-MBB/YHD796 film.** PMHS (24.0 mg, 0.400 mmol Si-H groups), MBB (76.0 mg, 0.254 mmol), 11UB (13.6 mg, 0.033 mmol), AZO46 (30.0 mg, 0.085 mmol) and YHD796 (0.7 mg, 0.5 wt%) were dissolved in 2 ml of toluene. The mixture was cast into a PTFE rectangular mould (2.0 cm long × 2.0 cm wide × 1.5 cm deep) after high-power ultrasonication for 5 min. After adding 80 μl of the pre-prepared Pt-catalyst solution (0.025 g of Pt(COD)Cl$_2$ was dissolved in 20 ml of CH$_2$Cl$_2$) into the above mixture, the PTFE mould was ultrasonicated for 5 min, to remove the bubbles in the mixture solution, and then heated in an oven at 60 °C for 4 h, to accomplish the first cross-linking stage. After cooling to room temperature, the LCE sample was carefully removed from the PTFE mould with the help of hexanes and then immediately cut into a strip (2.0 cm long × 0.2 cm wide, the thickness was ca. 0.1 mm), which was uniaxially stretched to ca. 140 ∼ 160% of the original length, fixed by using tapes and dried at room temperature.

**Preparation of the bilayer LCE ribbons.** The pre-crosslinked PMHS-AZO46-MBB/YHD796 composite film was obliquely placed on the top of the pre-cross-linked PMHS-MBB/YHD796 composite film with a crossed angle of either 45° or −45°. The fixed dual-layer LCE sample was heated at 60 °C in an oven for 72 h, to accomplish the full cross-linking purpose. Finally, the dual-layer film was further trimmed along the stretching direction of the top layer, into ribbons with a dimension of ca. 2.0 cm long × 0.2 cm wide × 0.2 mm thick.

**Data Availability.** Data supporting the findings of this study are available within the article (and its Supplementary Information files) and from the corresponding author upon request.

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

## Acknowledgements

This research was supported by National Natural Science Foundation of China (Grant Number 21374016) and Priority Academic Program Development of Jiangsu Higher Education Institutions.

## Author contributions

H.Y. designed this project. M.W. performed all the experiments. M.W., B.-P.L. and H.Y. analysed the data. M.W. and H.Y. wrote the manuscript.

## Additional information

**Competing financial interests:** The authors declare no competing financial interests.

