## [Peer Review File · Nature Communications]

Reviewers' Comments:

Reviewer #1 (Remarks to the Author)

The manuscript presents an approach of producing different types of mechanical actuation in elastomers by means of different external stimuli. This contribution is original as, to the best of my knowledge, the demonstrated modes of deformation have been only achieved in separate systems but not within the same sample. The manuscript is generally well written, although writing can be further improved. For example, the introduction could be made with more smooth transitions between paragraphs, there are a number of typos, and construction of some sentences is not very clear. I believe the conclusions are justified but would like to see more characterization, not just pictures. For example, how fast these different responses occur. This type of data would be rather easy to extract from movies and would be valuable for the manuscript, so I highly recommend it. Also, the paper would benefit from short explanations of physical underpinnings involved in activating different responses to external stimuli. It is true that they have been discussed in the papers that authors refers for separate realizations of different modes, but the discussion in this work would benefit from telling why and how rather than just what is accomplished. The referencing seems to be good and all current references adequately support the discussion in the manuscript, although the recent advances in controlling shapes of non responsive polymer objects/particles in liquid crystal perhaps should be discussed and referenced because of containing both bend and twist deformation of tube-like objects shaped into complex particles (see e.g. nature materials 13,258, 2014). The following statement "Further development of a smart soft actuator material with tunable two helical curling motions (left-handed vs right-handed) under two different stimuli (UV vs NIR), derived from this multi-layer, multi-composition strategy are under investigation" should not be part of the conclusion but could be part of the discussion before conclusions (there is no conclusion in it!). Overall, I believe the work will attract interest of a broad readership, is novel and can be published after the above remarks are adequately addressed.

Reviewer #2 (Remarks to the Author)

The authors report on a polysiloxane-based liquid crystal (LC) soft actuator capable of twisting and bending through modulation of the wavelength band of light stimuli (UV vs NIR). This is done fabricating a polymer bilayer. The top layer is a uniaxial aligned LC polymer film that contains another uniaxial bottom layer at a 45 angle in the middle of the film. Upon incorporation of azobenzene and IR dyes in the polymer layers two different motion modes were obtained. Although this is the first example of a single soft actuator that is able to bend and twist I do not recommend publication in Nature Communications. The (bilayer) method for obtaining bending and twisting is not novel and is based on cut and paste. Furthermore twisting and bending is obtained at different location of the polymer film.

Other issues:

- 1) It is unclear why upon exposure to UV light the middle part of the polymer is not twisting. What is observed if the film is turned upside down and exposed to UV light. The videos do not show that bending is reversible.
- 2) Only one polymer layer contains a crosslinker. The authors should explain why.
- 3) Upon exposure to IR light only twisting takes place at the edge of the bottom layer.

Reviewer #3 (Remarks to the Author)

This manuscript entitled "A plant tendril mimic soft actuator with phototunable bending and chiral twisting motion modes" by Professor Yang et al. presents the fabrication of polysiloxane based liquid crystal soft actuators with double layer architecture and their light-driven shape morphing phenomena. The authors have introduced a new method of fabricating covalently linked bilayer

architecture of liquid crystal elastomers in contrast to the usually used non-bonded (glued) bilayers. By judicious selection of materials for the top and bottom layers and their relative orientations, the authors have succeeded in prescribing bending and twisting motion modes to the double layer films. Reversibility of the light-driven actuation modes of the films is another promising feature of this research work. Furthermore, a noteworthy aspect of the research work is the selective photo addressability of the films by UV and NIR light irradiation. The results disclosed here on materials capable of performing two different reversible 3D shape modulations under irradiation with light of different energies would have far reaching impact in the elaboration of stimuli-responsive dynamic shape morphing materials and systems. Overall, this work is very interesting, and the experimental data are very solid. I would recommend this manuscript to be accepted for publication after making some minor revisions.

There are some typos, e.g., in line 57, the term "twist-Nematic" should be changed into "twist-nematic".

The authors may consider citing the following references related to photo-deformable and stimuli-responsive 3D liquid crystals. Intelligent Stimuli Responsive Materials: From Well-defined Nanostructures to Applications, Q. Li, Ed., John Wiley & Sons, New Jersey, 2013; Adv. Funct. Mater. 2016, 26, 10-28; Acc. Chem. Res. 2014, 47, 3184-3195; and Nature 2016, 531, 352-356.

Answers to comments from reviewers:

Reviewer #1:

1. Original comment: “The introduction could be made with more smooth transitions between paragraphs, there are a number of typos, and construction of some sentences is not very clear.”

Response: We thank the reviewer for this suggestion. We have revised the introduction part with more smooth transitions between paragraphs. All the changes are marked in red.

2. Original comment: “I believe the conclusions are justified but would like to see more characterization, not just pictures. For example, how fast these different responses occur. This type of data would be rather easy to extract from movies and would be valuable for the manuscript, so I highly recommend it.”

Response: We thank the reviewer very much for this suggestion. We have measured and recorded the included angle α vs UV illumination time diagrams (Fig. 3d), and the curvature ($1/r$) vs UV illumination time diagrams (Fig. 3e) of the two bilayer LCE ribbons to show the bending rates of two samples under the irradiation of UV light. We also added the temperature vs NIR illumination time diagrams to study the twisting behavior of two bilayer LCE ribbons, as shown in Fig. 4c-d. The discussions related to these data are also included in the revised manuscript (marked in red).

3. Original comments: “Also, the paper would benefit from short explanations of physical underpinnings involved in activating different responses to external stimuli. It is true that they have been discussed in the papers that authors refers for separate realizations of different modes, but the discussion in this work would benefit from telling why and how rather than just what is accomplished.”

Response: We thank the reviewer very much for this suggestion. We have added more explanations (marked in red) about the achievements of different modes in the revised manuscript. For example, “(Page 9) It is well known that the molar absorption coefficient of azobenzene is very high under 365 nm UV-light. The absorption of UV

photons promote electrons from the highest occupied molecular orbital (π -orbital) to the lowest unoccupied molecular orbital (π^* -orbital) of the azobenzene group, resulting in the isomerization of azobenzene from its thermodynamically stable *trans*-conformation to *cis*-conformation and the decrease of azobenzene's molecule length from ca. 10 Å to 5.6 Å.....(Page 10) the *cis-trans* isomerization extent of azobenzene chromophores would vary depending on the UV-penetration depth so that the top surface region of LCE sample would shrink much more than the bottom region. These different contraction ratios of the surface and bottom sides of LCE sample eventually forced the macroscopic material to bend towards UV source. When the film was turned upside down and exposed to UV light, the top non-azo-layer (**PMHS-MBB/YHD796** layer) would prevent UV photons from reaching the bottom azo-layer (**PMHS-AZO46-MBB/YHD796** layer) because of the limited penetration ability of UV light. Consequently, the two-layer-overlapped region of the LCE film kept motionless..... (Page 11) Consequently, the two layers tended to shrink along their own alignment orientations which were tilted to each other with a 45° or -45° angle, as illustrated in Supplementary Fig. 16. Such a non-uniform shrinkages created an incompatibility in the two-layer-overlapped region. Along each of their own alignment directions, the top and bottom layers had different contraction ratios, which resulted in two bending tendencies along the two alignment directions. The vector sum of the two bending deformations made an inclined angle with the long axis of the LCE ribbon, and eventually generated a chiral twisting power to force the macroscopically flat ribbon to curl in either right-handed or left-handed manner.”

4. Original comments: “The referencing seems to be good and all current references adequately support the discussion in the manuscript, although the recent advances in controlling shapes of non-responsive polymer objects/particles in liquid crystal perhaps should be discussed and referenced because of containing both bend and twist deformation of tube-like objects shaped into complex particles (see e.g. nature materials 13,258, 2014).”

Response: We thank the reviewer for this suggestion. We have cited this paper in Ref. 5, and given a short description about this beautiful work.

5. Original comments: “The following statement "Further development of a smart soft actuator material with tunable two helical curling motions (left-handed vs right-handed) under two different stimuli (UV vs NIR), derived from this multilayer, multi-composition strategy are under investigation" should not be part of the conclusion but could be part of the discussion before conclusions (there is no conclusion in it!).”

Response: We thank the reviewer for this suggestion. We have removed this statement from the conclusion part.

Reviewer #2:

1. Original comments: “Although this is the first example of a single soft actuator that is able to bend and twist I do not recommend publication in Nature Communications. The (bilayer) method for obtaining bending and twisting is not novel and is based on cut and paste. Furthermore twisting and bending is obtained at different location of the polymer film.”

Response: We thank the reviewer very much for giving our work a credit that is the first example of a single soft actuator being able to bend and twist.

All the previous examples with twisting motion mode, could be divided into two basic preparation strategies: monolayer with periodic patterns (ref. 32-34) or multilayer (including bilayer) structures (ref. 17,28-30). Although our system is a bilayer setup, our work has two key novelty points: (1) We develop a bilayer, dual-composition strategy, different from all the previous bilayer, mono-composition systems. This dual-composition idea is the essential point to realize bending and twisting under different stimuli. (2) Taking advantage of Finkelmann’s two-step crosslinking mechanism, the two different pre-crosslinked LCE samples could be spontaneously glued together during the second hydrosilylation crosslinking period,

without using extra adhesives which were indispensable in literature protocol (ref.17). In the revised manuscript, we further emphasize and explain this point: “The fixed dual-layer LCE film was heated at 60 °C in an oven for 72 h to accomplish the full crosslinking procedure, during which the leftover unreacted vinyl groups and Si-H groups on the interfaces of the two pre-crosslinked LCE samples would be covalently bonded together and consequently the two LCE layers would be spontaneously glued together.” In Discussion section, we further discuss about the two key novelty points of this work.

As can be seen in Fig. 3 and Supplementary Movie 1,2, the middle regions of two LCE ribbons containing both the top and bottom layers did also bend, which meant that twisting and bending could be obtained at the same location of the polymer film.

2. Original comments: “It is unclear why upon exposure to UV light the middle part of the polymer is not twisting. What is observed if the film is turned upside down and exposed to UV light. The videos do not show that bending is reversible.”

Response: We thank the reviewer for this critical point. The *cis-trans* isomerization extent of azobenzene chromophores vary dramatically depending on the UV-penetration depth, and UV penetration ability is much weaker than IR penetration. In such case, UV light might not be able to penetrate through the top layer so that the bottom side of the top layer would have very tiny or even no shrinkage. Meanwhile, the bottom layer has no azobenzene chromophores and no response to UV light. Thus, the overall sample would bend instead of curl towards UV.

When the LCE ribbon was turned upside down and exposed to UV light (new Fig. 3c and Supplementary Movie 3), the bending behavior took place at the two edges of the film, while the bilayer overlapped region remained motionless since the top non-azo-layer (**PMHS-MBB/YHD796** layer) would prevent UV photons from reaching the bottom azo-layer (**PMHS-AZO46-MBB/YHD796** layer) because of the limited penetration ability of UV light.

In the revised manuscript, the UV-responsive movies (Supplementary Movie 1,2) have been re-recorded to show that the bilayer LCE films could not only bend towards UV light, but also perform reversible unbending deformations under visible light irradiation. Furthermore, a new Supplementary Movie 3 has been recorded to present a side view of an upside down bilayer LCE ribbon under UV (365 nm) illumination.

3. Original comments: “Only one polymer layer contains a crosslinker. The authors should explain why.”

Response: We thank the reviewer for this comment. The top layer and the bottom layer both contain the crosslinker **11UB**, as shown in Fig. 1 and the Methods Section.

4. Original comments: “Upon exposure to IR light only twisting takes place at the edge of the bottom layer.”

Response: We thank the reviewer for this critical point. Upon exposure to IR light, twisting takes place at the overlapped regions of the top and bottom layers. Actually, the strategy of this “proof-of-idea” work could be further used to develop a variety of smart materials. For example, as schematically illustrated in the figure below, we could use a large mold to prepare a bottom layer with the same size of the top layer and stick them together. We can imagine that, such a sample would helically curl the whole ribbon under IR irradiation as the reviewer wished, and bend towards UV incoming direction if the azo-layer was placed on the top. The only drawback is that if UV illuminated on the non-azo-layer, the whole LCE ribbon would have no response and keep motionless.

Figure 1. Schematic illustration of preparing a LCE ribbon with the same sized top and bottom layers.

Reviewer #3:

1. Original comments: “There are some typos, e.g., in line 57, the term “twist-Nematic” should be changed into “twist-nematic”.”

Response: We thank the reviewer for this suggestion. We have revised it.

2. Original comments: “The authors may consider citing the following references related to photo-deformable and stimuli-responsive 3D liquid crystals. Intelligent Stimuli Responsive Materials: From Well-defined Nanostructures to Applications, Q. Li, Ed., John Wiley & Sons, New Jersey, 2013; Adv. Funct. Mater. 2016, 26, 10-28; Acc. Chem. Res. 2014, 47, 3184-3195; and Nature 2016, 531, 352-356.”

Response: We thank the reviewer for this suggestion. We have cited these references in Ref.35-38.

Furthermore, we have cited some more references related to smart soft actuators and photoresponsive LCEs in Ref. 12,14,16,21-24,26,27,29,39,41,50. All the new references are marked in red.

We appreciate for Editors/Reviewers' warm work earnestly, and hope that this revised manuscript would be now suitable for publication. Thank you very much for your consideration.

Sincerely,
Hong Yang

Reviewers' comments:

Reviewer #1 (Remarks to the Author):

Authors accounted for my suggestions, such as the ones related to quantitative characterization of response times. I believe the manuscript is substantially improved and recommend publication

Reviewer #2 (Remarks to the Author):

I still have a problem with the current manuscript. The author should clearly show that bending and curling is not only possible at certain locations by this "cut and paste" method. The experiment proposed by the authors (Figure 1, response letter) should be carried out to show that an actuator can be made that curl or bend a whole ribbon under IR or UV irradiation, respectively. All other issues have been addressed.

Dear Dr. Matsiko:

Please find enclosed a revised version of our manuscript entitled “A Plant Tendril Mimic Soft Actuator with Phototunable Bending and Chiral Twisting Motion Modes” (ID: NCOMMS-16-14361B). Following the comments and suggestions made by the referees, we have substantially modified our manuscript. We would like to present our point-by-point answers to the comments made by the referees.

Answers to comments from reviewers:

Reviewer #2:

1. Original comment: “I still have a problem with the current manuscript. The author should clearly show that bending and curling is not only possible at certain locations by this “cut and paste” method. The experiment proposed by the authors (Figure 1, response letter) should be carried out to show that an actuator can be made that curl or bend a whole ribbon under IR or UV irradiation, respectively. All other issues have been addressed.”

Response: We thank the reviewer for this wonderful suggestion. As the reviewer suggested, we have prepared a new dual-layer dual-composition soft actuator, whose top layer was of the same size as the bottom layer. As shown in new Fig. 5 and Supplementary Movie 6,7, such a same-sized-bilayer sample did helically curl the whole ribbon under IR irradiation, and did bend the whole ribbon under UV irradiation.

We appreciate for Editors/Reviewers’ warm work earnestly, and hope that this revised manuscript would be now suitable for publication. Thank you very much for your consideration.

Sincerely,
Hong Yang

Reviewers' Comments:

Reviewer #2 (Remarks to the Author)

My last concern has been satisfactorily addressed. The manuscript is now suitable for publication.

Dear Dr. Saini:

Please find enclosed a revised version of our manuscript entitled “A Plant Tendril Mimic Soft Actuator with Phototunable Bending and Chiral Twisting Motion Modes” (ID: NCOMMS-16-14361C). Following the comments and suggestions made by the referees, we have substantially modified our manuscript. We would like to present our point-by-point answers to the comments made by the referees.

Answers to comments from reviewers:

Reviewer #2:

1. Original comment: “My last concern has been satisfactorily addressed. The manuscript is now suitable for publication.”

Response: We thank all the reviewers for the positive assessments of our revision. We also appreciate all the reviewers for the valuable suggestions on the manuscript. These suggestions, ranging from the data analysis to the technological writing, can help us improve the quality of the paper indeed. We think we benefit greatly by these suggestions.

We appreciate for Editors/Reviewers’ warm work earnestly, and hope that this revised manuscript would be now suitable for publication. Thank you very much for your consideration.

Sincerely,

Hong Yang